# Integration of evidence into Theory of Change frameworks in the healthcare sector: A rapid systematic review

Davi Mamblona Marques Romão[1]*, Cecilia Setti[1], Leo Heikiti Maeda Arruda[2], Roberta Crevelário de Melo[3], Bruna Carolina de Araujo[3], Audrey R. Tan[4], Peter Nichols DeMaio[5‡], Tanja Kuchenmüller[5‡]

1 Instituto Veredas, São Paulo, São Paulo, Brazil, 2 Doebem, São Paulo, São Paulo, Brazil, 3 Evidence Center, Instituto de Saúde, São Paulo, São Paulo, Brazil, 4 University College London, London, United Kingdom, 5 Evidence to Policy and Impact, World Health Organization, Geneva, Switzerland

☯ These authors contributed equally to this work.
‡ PND and TK also contributed equally to this work.
* davi@veredas.org

**Data Availability Statement:** All relevant data are within the manuscript and its Supporting information files.

## Abstract

### Background

Theory of Change (ToC) has become an established approach to design and evaluate interventions. While ToC should—in line with the growing international focus on evidence-informed health decision-making–consider explicit approaches to incorporate evidence, there is limited guidance on how this should be done. This rapid review aims to identify and synthesize the available literature on how to systematically use research evidence when developing or adapting ToCs in the health sector.

### Methods

A rapid review methodology using a systematic approach, was designed. Eight electronic databases were consulted to search for peer-reviewed and gray publications detailing tools, methods, and recommendations promoting the systematic integration of research evidence in ToCs. The included studies were compared, and the findings summarized qualitatively into themes to identify key principles, stages, and procedures, guiding the systematic integration of research evidence when developing or revising a ToC.

### Results

This review included 18 studies. The main sources from which evidence was retrieved in the ToC development process were institutional data, literature searches, and stakeholder consultation. There was a variety of ways of finding and using evidence in ToC. Firstly, the review provided an overview of existing definitions of ToC, methods applied in ToC development and the related ToC stages. Secondly, a typology of 7 stages relevant for evidence integration into ToCs was developed, outlining the types of evidence and research methods the included studies applied for each of the proposed stages.

**Funding:** This project was funded by the World Health Organization (https://www.who.int). The authors declare no known financial conflicts of interest. TK is employed full-time with WHO, working in the fields of research, evidence and policy. PND had consultancy contracts with WHO when the work on the Theory of Change was commissioned. The authors alone are responsible for the views expressed in this paper and they do not necessarily represent the views, decisions or policies of the institutions with which they are affiliated. The funders had no role in study design, data collection and analysis, decision to publish, or preparation of the manuscript.

**Competing interests:** The authors have declared that no competing interests exist.

## Conclusion

This rapid review adds to the existing literature in two ways. First, it provides an up-to-date and comprehensive review of the existing methods for incorporating evidence into ToC development in the health sector. Second, it offers a new typology guiding any future endeavors of incorporating evidence into ToCs.

## Background

### What is a Theory of Change?

A Theory of Change (ToC) comprises a sequence of causal steps that describe how an intervention is expected to produce certain outputs and outcomes. These steps often include what inputs will be used, which activities will be conducted, what outputs will be produced, and, finally, what outcomes are expected to be achieved through this process, i.e, what "change" is expected to happen [1, 2]. According to the Aspen Institute [3], Theory of Change (ToC) can be understood as a tool to facilitate the development of solutions to complex social problems. A ToC is generally presented in graphic form, as a diagram, showing the connections between interventions and outcomes (causal pathways) while explicitly stating assumptions and related evidence [4]. This diagram usually represents a working model containing preconditions, expected results, rationales, assumptions, and indicators.

The concept of ToC was preceded, during the 50s and 60s, by an approach called Program Theory, which proposed a way of understanding and planning interventions with focus on the roles of context, input, processes, and products for intervention design [5]. Over time, increasingly more attention was given to explaining how an intervention was expected to work in terms of its underlying causal links [6]. In 1995, The Roundtable on Comprehensive Community Development, supported by the Aspen Institute [7], advocated for the importance of making it clear for stakeholders the assumptions of how change is to be achieved by a program, popularizing the term Theory of Change, described as "a theory of how and why an initiative works" [8].

The concept of ToC is related to, and often overlaps with, several other tools and frameworks, such as Logical Model, Program Theory, Action Theory or Logical Framework. A Logical Model, for instance, is defined as a graphical representation of a program theory, showing the relevant causal connections up to the results [5]. A Program Theory is an explicit theory of how an intervention is expected to achieve its goals [5]. An Action Theory, on the other hand, describes the pathways for developing an intervention. Finally, a Logical Framework, or Logframe, is a method for planning and evaluation, usually specifying inputs, activities, outputs, outcomes, and impacts in a matrix, and possibly also including indicators and assumptions [9, 10]. Logframes are considered less flexible than ToC diagrams, and more limited in representing multiple, interacting, and nonlinear causal pathways. Importantly, ToCs should make assumptions and contextual factors explicit to explain the causal steps, i.e. why and how an intervention is expected to lead to the anticipated changes [9]. Furthermore, a core feature that distinguishes ToC from other similar approaches is the emphasis on it being a living process, constantly updated based on lessons learned [11, 12].

These steps often include what inputs will be used, what activities will be conducted, what outputs will be produced, and, finally, what outcomes are expected to be achieved through this process. Fig 1 below presents a model of how a ToC can be depicted, including how to include

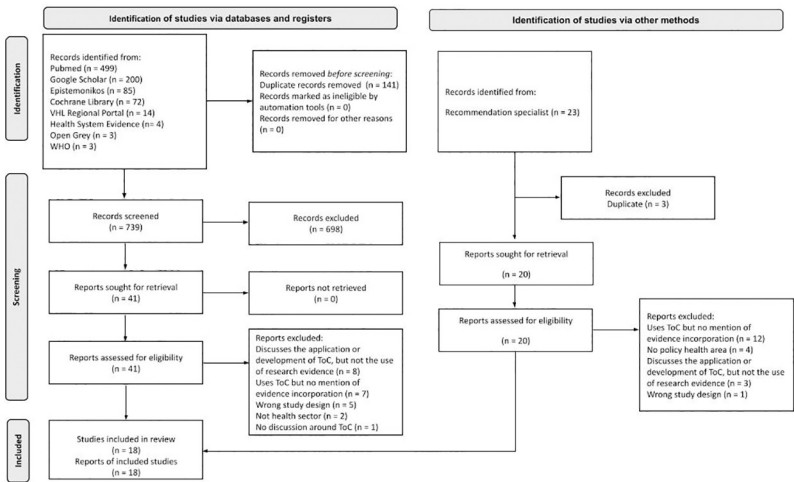

**Fig 1. Example of Theory of Change framework.** Source: De Silva et al., 2014, p.4 [13].

assumptions and indicators. The following Fig 2 illustrates a concrete ToC for interventions addressing Domestic Violence (DV) against pregnant women:

In this review, Theory of Change is defined, according to De Silva et al. (2014b, p.2) [15], as follows:

> "ToC is 'a theory of how and why an initiative works' which can be empirically tested by measuring indicators for every expected step on the hypothesized causal pathway to impact. It is developed in collaboration with stakeholders and modified throughout the intervention development and evaluation process through an 'ongoing process of reflection to explore change and how it happens'. It is visually represented in a ToC map which is a graphic representation of the causal pathways through which an intervention is expected to achieve its impact within the constraints of the setting in which it is implemented".

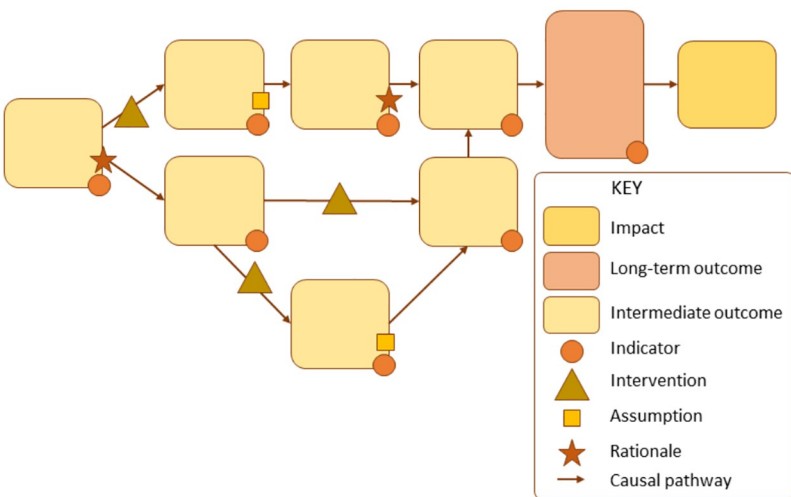

**Fig 2. Example of theory of change for an intervention.** Source: Sapkota et al., 2019, p.6 [14].

Visually, each intervention is connected to an expected result, in a causal structure, in order to reveal the necessary activities to promote change [16] and without losing the implicit complexity of the actions. Moreover, the connections between inputs, outputs, and outcomes are explained to demonstrate how and why certain actions are expected to lead to change [16].

ToC can be used for many purposes. It serves as a tool to plan an intervention since it makes all steps and assumptions explicit and visible. It can also be used as a tool for communicating and discussing the rationale of the intervention with staff members, stakeholders, the target population, and other audiences. Therefore, a ToC can be useful for providing transparency about why certain actions have been taken and allows for the intervention to be better replicated in the future, facilitating coordination and collaboration. Finally, given its ability to depict the causal process through which an intervention is meant to work and what change it is expected to produce, ToC can also inform monitoring and evaluation processes. In depicting the key steps and expected deliverables, it facilitates the identification and development of key indicators [9, 16, 17].

While the ToC approach is most commonly used as an intervention planning and design framework, it can also be used after an intervention has been implemented as a way of synthesizing empirical evidence of a pilot study [18], a single intervention [18], or a collection of studies [14, 19–22].

By articulating long-term results, pre-conditions, and interventions, the ToC provides a basis for implementing and documenting policies, strategic plans, and monitoring and evaluation processes [16].

ToC has been widely used in the health sector for planning, depicting, implementing, and evaluating interventions [11, 12, 23, 24]. For instance, Barnhart et al. (2020) [18] developed a ToC based on the experience of three different implementation phases of a maternal and child health program, the Better Birth program [18]. Barriers, lessons learned and ways of applying these lessons were identified, and later integrated into a renewed ToC. In another example, Aggarwal (2021) [19] reviewed the literature for potential interventions and delivery mechanisms for reducing the self-harm of individuals in countries of low and middle income [19]. Based on the findings of the review, the authors propose a ToC for a model of delivery of psychosocial interventions. Furthermore, in the field of Implementation Science, ToC has been a recurring choice to describe models, plan solutions to complex problems and evaluate programs in the health area, as described by Breuer et al. (2016) [23]. Recent reviews suggest there is a growing number of published evaluations in the health area that use ToC, as identified by Breuer et al. (2016) [23] and Lam et al. (2021) [24]. However, both reviews highlight that studies, overall, lack an appropriate level of detail regarding how the ToCs were developed and used.

## The use of evidence in Theory of Change frameworks

Evidence-informed decision-making has been increasingly used as a method in clinical policy and practice. The use of evidence makes the decision-making processes more explicit, favoring transparency, reproducibility, credibility, and reliability [25]. We argue that the development and use of ToC frameworks should, therefore, also include explicit and systematic considerations on how to incorporate evidence.

In this review, we will follow the Lomas et al. (2005) [26] conceptualization of evidence as "facts (actual or asserted) intended for use in support of a conclusion" (p.1). The authors [26] distinguish colloquial evidence (or tacit knowledge), provided by stakeholders' opinions and viewpoints, from scientific evidence (or explicit knowledge), gathered by a prescribed set of procedures in order to produce knowledge recognized as explicit, systemic, and replicable, and

further distinguishes two scientific views on evidence depending on whether they emphasize context-free general truths (more aligned with evidence-based medicine), or context-sensitive facts (more aligned with the applied social sciences). The authors also highlight that combining and interpreting these forms of evidence requires a deliberative process, often involving the participation of stakeholders to judge the relative weight of different pieces of evidence.

In theory, a ToC would be based on the analysis of available scientific evidence (explicit knowledge) and the consultation and deliberation with stakeholders (tacit or colloquial knowledge) [2] developed through rigorous and participatory processes. ToC approaches, however, usually emphasize stakeholder input (stakeholder consultation in the development of a plan to articulate the initial conditions, objectives, and means to achieve the desired results in a causal structure [16]) while making limited use of explicit or academic knowledge such as program evaluations and published research [23].

Two literature reviews have previously investigated the methods used to incorporate evidence into ToCs in the health sector. In a systematic review, Breuer et al. (2016) [23] investigated how ToCs are used to design and evaluate public health interventions. The authors identified the following methods used to incorporate evidence into ToC frameworks: "workshops and working groups, document reviews, interviews and discussions, surveys, program observation, literature reviews, and existing conceptual frameworks or theory, [...] consultations or interviews [with stakeholders]" (p.4). Moreover, in a scoping review, Lam et al. (2021) [24] investigated how ToCs are used in food security contexts. The authors find that the methods applied for using evidence in ToCs were primarily participatory approaches, such as workshops or interviews, i. e., favoring tacit knowledge. They also report that most studies do not provide details on how the ToCs were developed, which is "consistent with the sentiments in the literature, which highlighted the limited guidance, training, and agreement on how to develop a ToC" (p.7) [24].

In this review, we expand the existing literature by focusing on how to systematically use research evidence to inform the development of ToCs in the health sector. Consistent with the previous reviews, our findings below show that almost none of the recent studies have systematically investigated the use of evidence to inform ToCs in the health sector. Furthermore, we did not encounter any study that provides a comprehensive and rigorous account on what methods can or should be used to incorporate evidence into ToCs. Most of the primary studies that address how a ToC was developed only give brief descriptions of the methods applied to incorporate evidence into this process. These methods include, for instance, structured stakeholder consultation, literature searches and the use of institutional data, which are applied as sources of information to draft or improve ToCs. However, each study seemed to choose specific versions of these methods (e.g. stakeholder consultations) without a clear rationale for why this choice was made over the alternatives (e.g. a literature review).

This project was commissioned by the World Health Organization, which identified the need for more explicit use of evidence in ToC development. This rapid literature review specifically aims to advance our knowledge on how to enhance the systematic and transparent use of the best available evidence in the ToC development for decision-making in the health sector. The rapid review methodology was chosen to ensure agility and efficiency in applying the available resources while keeping methodological rigor.

This rapid review adds to the existing literature in two ways. First, it provides an up-to-date and comprehensive review of the existing methods for systematically incorporating evidence into ToC frameworks in the health sector. Second, it creates a typology to classify the findings from the literature. In this typology, we propose a sequence of 7 stages relevant to the process of evidence incorporation into ToCs. We also indicate which research methods the included studies applied for each of the proposed stages. These 7 stages were developed through a

**Table 1. PCCS acronym for the research question.**

| P | Problem | Lack of knowledge on how evidence is used in ToC frameworks in the health sector |
|---|---|---|
| C | Condition | Use of evidence in ToC |
| C | Context | ToC in the health sector |
| S | Study design | Systematic reviews; Guides and methodological studies |

Source: authors' elaboration.

qualitative analysis of the data extracted from the included studies (further details provided in the "Data Synthesis" section below). Each stage corresponds to a specific set of objectives. The stages are: 1) "Define the problem", 2) "Define expected outcomes", 3) "Define interventions", 4) "Define change mechanisms", 5) "Model ToC", 6) "Validate ToC" and 7) "Revise ToC".

## Objectives

This rapid review aims to identify and synthesize the available literature on how to systematically use research evidence to inform the development and revision of Theories of Change in the health sector.

### Research questions

To that end, we will address the following key questions:

1. What are the available tools, methods, and recommendations to systematically incorporate research evidence into the development and adaptation of Theories of Change in the health sector?

2. What are the main principles, processes, and practical steps recommended by the literature on how to systematically incorporate research evidence into the development and adaptation of theories of change in the health sector?

To answer these questions, the acronym PCCS was used, as follows (Table 1):

## Materials and methods

### Study design

This project applied a rapid review methodology. Rapid reviews can be defined as a type of literature review in which "the steps of the systematic review are streamlined or accelerated to produce evidence in a shortened timeframe." (p. XIII) [27]. In a rapid review, evidence is synthesized and its validity is assessed using an abbreviated systematic review method, so to identify results in a shorter timeframe. Regardless of the abbreviated approach, this endeavor followed the principles of the scientific method and the key principles of evidence synthesis, including rigor, transparency, and reproducibility. This rapid review was conducted and reported following the process outlined by Tricco et al. (2017) [27]. We used a rapid review approach to be able to provide high quality evidence for decision making, in a timely way. Given the time and budget limitations of this project, some adaptations were made to adjust to the needs of the rapid review design. These adaptations are described in the section "Shortcuts taken for the rapid response" below.

The study protocol was published in the Open Science Framework (OSF) platform (register: osf.io/t53sm) [28]. After the publication of the protocol, we made an addition in the research design by expanding the inclusion criteria also to include scoping reviews.

## Eligibility criteria

The inclusion criteria applied in this review were:

- **Topic**: studies that investigated how research evidence was used in the development and adaptation of ToC in health policies; or studies that explicitly described how evidence was or should be used to inform the development or adaptation of the ToC in the health sector;

- **Study type**: systematic reviews, scoping reviews, guides, or methodological studies. The methodological studies could be case studies, qualitative or quantitative studies, as long as their primary focus was how to develop or update a ToC. We excluded empirical primary studies, overviews, integrative reviews, evidence syntheses, technology assessments, and economic assessments;

- **Language**: published in English, Spanish, or Portuguese;

- **Date**: published on any date;

- **Place**: any country or region.

## Shortcuts taken for the rapid review

In this rapid review, only the screening process was done by two reviewers independently. Data extraction and methodological quality assessment of the studies were done by one researcher and reviewed by a second.

## Data sources and searches

Published articles were searched on November 8, 2021, in the following databases: PubMed, BVSalud, Cochrane Library, Health Systems Evidence (HSE), Health Evidence, and Epistemonikos.

Gray literature was retrieved from OpenGrey and Google Scholar search engines limited to the first ten pages. A list of references obtained from a WHO researcher that had been previously investigating the topic was also screened. Due to time constraints, no additional search for gray literature was conducted.

Search strategies were developed based on the combination of keywords structured by the acronym PCCS, using the MeSH search terms on the PubMed database. The strategy was adapted to the other databases.

Additional studies were retrieved from a list provided by a topic expert and the WHO library.

The implemented searches can be found in S1 Appendix.

## Study selection and data screening

Study screening was preceded by calibration of the inclusion and exclusion criteria among all reviewers using a sample of the search results. After calibration, two reviewers selected the studies through title and abstract screening independently. Disagreements were decided by a third reviewer.

After title and abstract screening, two researchers independently screened the included studies in full-text. Conflicts were resolved through consensus or by a third reviewer.

The screening was conducted in the reference management software, Rayann QCRI [29].

## Missing data

Only reports published in Portuguese, Spanish, and English were included, and potentially relevant information in other languages was not considered. In addition, primary studies that addressed the use of ToC to apply evidence in the health sector but that were not included in systematic reviews or scoping reviews were also not covered by this study. Finally, relevant literature from other fields than health might also have had relevant information to answer this review's question, but they were not included.

## Data extraction

Data extraction was conducted according to the research questions. A data extraction spreadsheet was made, observing the main elements to inform this rapid review, and covered the following categories: first author, published year, study objective, population and sample characteristics, method, main findings (divided into Definition of ToC, Participants in the ToC development, Steps for the development of a ToC, Steps for revising a Theory of Change, Use of research evidence in the development of a ToC, and Use of research evidence for revising a ToC), study limitations, conclusions, conflicts of interest and last year of search. In the subcategories "Use of research evidence in the development of a ToC" and "Use of research evidence for revising a ToC", we captured all relevant information from the primary studies regarding their reported strategies for evidence incorporation into ToCs. Data for the other subcategories were collected to identify and contextualize each study and provide further information on how ToCs were used and how evidence was considered. The fields "Definition of ToC", "Steps for the development of a ToC" and "Steps for revising a Theory of Change" were added to the data extraction template after a round of initial piloting of the data extraction process to facilitate the interpretation of the research findings.

## Quality assessment

Quality assessment was done by one researcher and reviewed by another. Systematic reviews were assessed using the AMSTAR 2 tool [30]. Systematic reviews were classified according to their confidence degrees as high, moderate, low, or critically low. This classification followed the AMSTAR 2 critical domains that check for the presence of a search protocol, search strategy, inclusion and exclusion criteria, assessment of the risk of bias, and interpretation of results. To assess the quality of the scoping review, we used the Joanna Briggs Institute's Checklist for Systematic Reviews [31]. To evaluate the studies self-described as methodological, we used the Joanna Briggs Institute's Checklist for Qualitative Studies [31]. For the other included studies, that we also considered methodological, we applied quality assessment tools compatible with each of their specific designs, they were: Joanna Briggs Institute's Checklist for Qualitative Studies [31] (4 studies), Joanna Briggs Institute's Checklist for Text and Opinion [31] (2 studies), Scale for the Assessment of Narrative Review Articles (SANRA) [32] (1 study), and Critical Appraisal of a Case Study tool from the Center for Evidence-Based Management [33] (1 study). The quality assessment results can be found in S2 Appendix.

## Data synthesis

After data extraction, studies were compared, and the findings were summarized qualitatively into themes, in the form of key principles, steps, and procedures for the systematic incorporation of research evidence into the development and adaptation of a ToC. A qualitative narrative synthesis was conducted regarding "Definitions of ToC", "Participants in the ToC development", "Methods for the development of the ToC" and "Use of research evidence in the development and revision of a ToC".

To further contextualize how and when evidence can be integrated into the ToC development process, we also identified, based on the available literature, the main stages relevant for the process of evidence incorporation into ToCs. The basis for this process was the extracted data regarding the steps for developing and revising a ToC and the strategies for evidence use (see S6 Appendix for the relevant extracted data). Through an inductive qualitative data analysis process, we created a typology of stages and coded the extracted information accordingly. When grouping the steps based on content similarity, we realized they could be classified with labels similar to the phases often included in the "policy cycle", such as "Policy Formulation", "Implementation" and "Evaluation" (p.33) [34]. Thus we regrouped the data following this framework, adapting some of these phases as labels (i.e. "Define the problem", "Define interventions"), and creating new ones when necessary (i.e. "Define change mechanisms", "Model ToC", "Validate ToC", "Revise ToC") based on our content similarity analysis. Here, we present these labels as stages for the development of ToCs. The different stages reflect the objectives and functions of each step, as shown in section "Stages for the development of a ToC" below and in S3 Appendix. Thereafter, we conducted a frequency count to assess how many of the included studies reported each stage. For ease of understanding, these stages were organized in a logical sequence, according to our interpretation.

In addition to categorizing the stages for developing a ToC, we classified the extracted data in terms of "types of activities" conducted. This typology was also created through an inductive qualitative data analysis process, based on the original extracted material. Within each stage, we grouped the included methods according to the specific actions that were conducted. While the classification in stages reflects the goal or purpose of each step taken, the classification into the types of activities summarizes the concrete actions undertaken to achieve this goal (see section "Stages for the development of a ToC" below).

## Results

### Study inclusion

The search retrieved 880 records, of which 141 were removed as duplicates. 698 articles were excluded at the title and abstract screening as they did not meet the eligibility criteria. 41 documents were selected for full-text screening, of which 23 were excluded for not discussing ToC, the use of evidence, or not matching the inclusion criteria. In addition, we screened 23 articles recommended by an expert on the topic. After excluding duplicates, the remaining studies were screened in full-text. Based on the exclusion criteria, all of these studies were excluded. The excluded articles and the reasons for exclusion are listed in S4 Appendix, separated between articles found in the databases and those recommended by the expert (Fig 3). Finally, 18 remaining articles were included in this rapid review.

Despite there being no limitation on the year of publication as an inclusion criterion, all included studies (n = 18) were published between 2013 and 2021 [1, 13, 14, 18–24, 36–43], and among those, the mode (n = 5) was the year 2021 [19, 22, 24, 39, 42]. For more details on the characteristics of each study and the data extracted, see S5 and S6 Appendices.

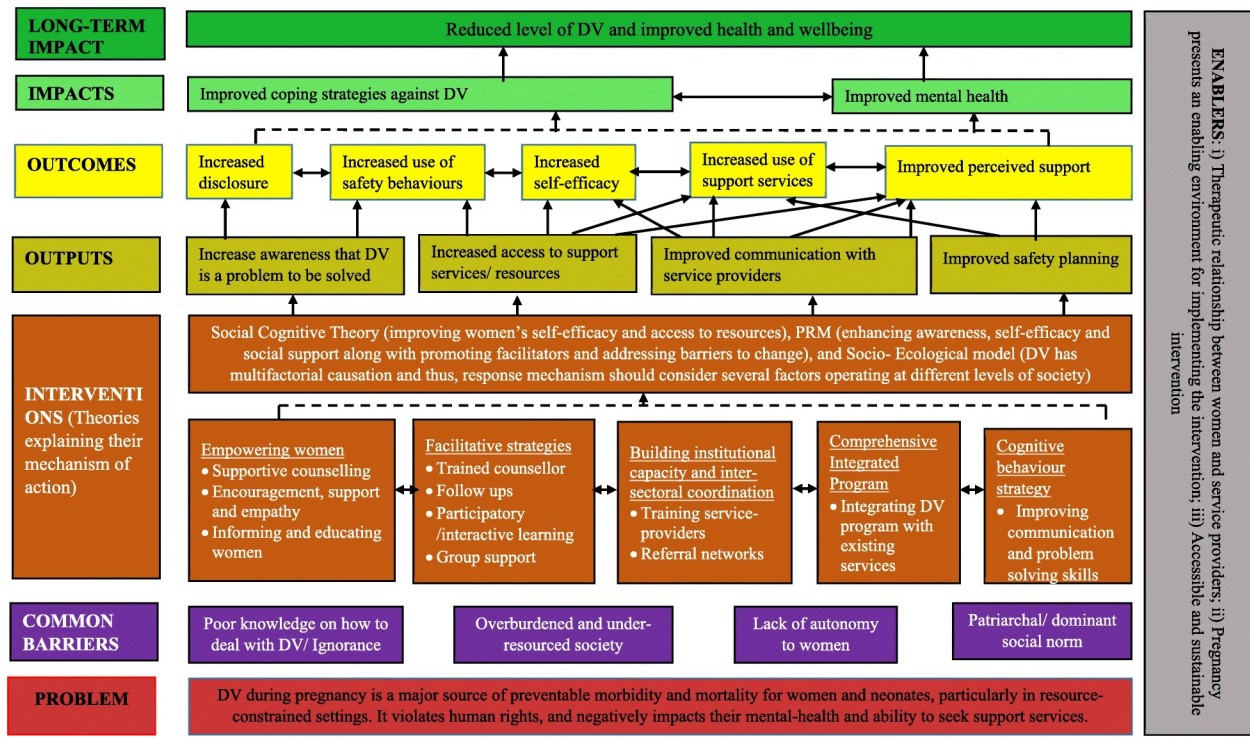

**Fig 3. PRISMA flowchart.** Source: authors' elaboration, adapted from PRISMA 2020 [35].

## Characteristics of included studies

The studies presented different designs with seven systematic reviews [19, 21–23], one scoping review [24], and ten studies with other designs that were considered "methodological" since their primary focus was on how to develop or update a ToC. The included methodological studies had a variety of study methods, as reported by their respective authors: case study (n = 1) [18], mixed methods (qualitative and quantitative) (n = 1) [42], methodology studies (n = 2) [36, 37], report (n = 1) [40], formative research (n = 1) [13]. Four publications [1, 20, 38, 43] did not name the study design. They were classified by the authors of this rapid review as methodological studies based on the information described in the method section, for the purpose of quality assessment.

Regarding the countries conducting the studies, five studies were carried out in countries that, according to The World Bank (2021) [44], are classified as high-income economies (HIE) [21, 23, 36, 37, 42], one in Lower-middle-income economies (LMIE) [14], and one in Upper-middle-income economies (UMIE) [43]. Two studies included both HIE and LMIE [18, 19]. The remaining studies did not report this information or it did not apply to the study design [13, 20, 22, 38–41]. Except for De Buck et al. (2018) [1], which had authors from both HIE and LMIE, the authors of the other 17 included studies were affiliated to institutions from HIE [13, 14, 18, 19, 22, 24, 36–43].

Sixteen studies reported no conflict of interest [1, 13, 14, 18–20, 24, 36, 39]. One study included information about funding but not on competing interests [40]. One study did not have information on either [20].

The included studies focused on the following themes and sectors: public health [13, 21, 23, 40], nutrition [24, 38], healthy eating and physical exercise [20], mental health [1, 22, 36],

health promotion [1], violence prevention [4], obstetrics [18], immunization data [39], research-policy gap [43], realist RCTs [37], and reducing sedentary behavior [42].

## Methodological quality of included studies

The methodological quality of seven systematic reviews was assessed using AMSTAR-2 [30]. Five of these were considered critically low [19, 21–23, 39] and two considered low [14, 41]. None of the publications provided a list of excluded studies with justifications [14, 19, 21–23, 39, 41]. In addition, the five critically low did not account for biases [10, 12, 13, 17, 19, 21–23], or did not refer to methods being established prior to the review [39]. Six studies were assessed with the JBI Critical Appraisal Checklist for Qualitative Research [31]. These studies contained two important weaknesses: no statement locating the researcher culturally or theoretically, nor any mention of approval by the ethics committee [1, 20, 36, 37, 42, 43]. One study [24] was assessed with the JBI Critical Appraisal Checklist for systematic reviews and research syntheses [31]. However, methods to mitigate potential errors in the findings in this study were not found. Two studies were assessed with the JBI Critical Appraisal Checklist for text and opinion papers [31], and the reports were considered satisfactory [13, 40]. One study [38] was assessed with SANRA [32] and had a final score of 10/12. The two limitations of the study were that the electronic search report was not detailed enough and the selection of studies was not well described. One case study was assessed with the CEBMA tool [33]. In this study, there is no mention of whether the data analysis was repeated by another researcher and it is not clear whether the researcher's perspective was taken into account [18]. More details are found in S2 Appendix.

## Definitions of ToC

Nine studies [1, 13, 14,18, 22, 23, 38, 40, 43] presented a definition of ToC, with some differences between them, as shown in Table 2.

## Participants in the ToC development

Four studies reported that their respective authors participated in the development of the ToC [22, 36, 39, 40]. Other studies included multidisciplinary groups composed of professionals [42], service users [20], program evaluators and implementers, community members [18], methodological experts, and experts on the topic [1]. Ten studies did not report any information about the participants in the development of the ToC [13, 14, 19, 21, 23, 24, 37, 38, 41, 43].

## Methods for the development of the ToC

Some of the studies [14, 19, 20] followed existing methods for intervention design in their efforts to build a ToC. The methods used were De Silva (2014) [13], the Six steps in quality intervention development (6SQuID) [45] framework [36, 42], and the guidelines [43] offered by the World Bank [46] and Morra and Rist [47].

Three studies [21, 22, 41] synthesized ToC evidence from previously published ToCs. Tancred et al. (2018) [41], for instance, conducted a systematic review to synthesize relevant elements for a ToC for school-based interventions integrating health and academic education to prevent violence and substance use among students. This review searched for other ToCs that were synthesized through a meta-ethnographic analysis.

**Table 2. Definitions of ToC from the included studies.**

| Study | ToC Definition |
|---|---|
| Barnhart et al. 2021 [18] | "The term theory of change (TOC) was popularized by Carol Weiss to describe a tool that defines and expresses researchers' underlying assumptions and hypotheses about the processes through which a complex intervention improves outcomes". |
| Breuer et al., 2016 [23] | "We define it as an approach which describes how a programme brings about specific long-term outcomes through a logical sequence of intermediate outcomes". |
| De Buck et al., 2018 [1] | "Weiss described a ToC as 'a theory of how and why an initiative works'. More specifically, it aims at developing a shared understanding of the processes and underlying mechanisms by which interventions are likely to work[…]". |
| De Silva et al, 2014 [13] | "Theory of Change (ToC) is an approach to developing, implementing and evaluating programmes of development, and has been applied across a wide range of programmatic contexts". |
| Mayne, 2015 [38] | "A theory of change adds to an impact pathways (IPs) by describing the causal assumptions behind the links in the pathways–what has to happen for the causal linkages to be realized […]". |
| Meiksin et al. 2021 [22] | "Intervention theories of change typically draw on existing scientific theories of behaviour (which consider factors that predict behaviours) and/or existing scientific theories of behaviour change (which propose general mechanisms of changing behaviour). Intervention theories of change make explicit the hypothesised mechanisms through which intervention activities are intended to generate outcomes, helping developers to systematically consider and describe which psychological, social or other factors interventions should address to achieve their intended outcomes. They also help evaluators determine what to measure to assess whether and how an intervention works and which components, if any, are most important." |
| Rippon et al., 2017 [40] | "Some people view it as a tool and methodology to map out the logical sequence of an initiative from inputs to outcomes. Other people see it as a deeper reflective process and dialogue amongst colleagues and stakeholders, reflecting on the values, world views and philosophies of change that make more explicit people's underlying assumptions of how and why change might happen as an outcome of the initiative[…]" |
| Sapkota et al., 2019 [14] | "ToC is a comprehensive description and illustration of how and why a desired change is expected to happen in a particular context. The development of a ToC is an iterative process and can use various methods including review of existing information, interviews and/or consultation with stakeholders, with the choice of the method being based upon what is locally feasible and acceptable. The use of ToC approach is widespread in the field of public health and many development organizations have evaluated this method as accessible, feasible, and useful". |
| Yearwood, 2018 [43] | "Theory of change (ToC), on the other hand, is a more flexible format. ToC describes how a program, through a logical sequence of intermediate outcomes, can bring about specific long-term outcomes. The pathways through which interventions work to achieve a desired impact are made explicit. ToC also allows for depiction of multiple pathways and feedback loops, which is more reflective of the complexities involved in real-world settings". |

Source: authors' elaboration.

## Stages for the development of a ToC

The included studies generally did not provide a detailed account of what steps were taken to develop their respective ToCs. However, from the available information, it is possible to see that the development of the ToC was rarely a linear or hierarchical process. In reality, the development process is continuous, involving reflection and adaptation as new evidence comes to light, requiring the causal pathways to be changed [15].

All studies included in this review presented at least one stage adopted for the construction of the ToC. In total, seven development stages were identified. The most frequent type was "Define interventions" [1, 19, 22, 36, 39, 41, 42] and "Model ToC" [13, 18–20, 22, 23, 38, 42, 43], mentioned in 55.5% (n = 10) of the included studies. All identified stages and their respective frequencies can be found in Table 3.

**Table 3. Frequency of each stage of ToC development.**

| Stages to develop a ToC | Frequency % (n) |
|---|---|
| **Define interventions** | 55.5 (10) [1, 18–22, 36, 39, 41, 43] |
| **Model ToC** | 55.5 (10) [13, 18–23, 38, 42, 43] |
| **Define change mechanisms** | 38.8 (7) [13, 14, 21, 36, 37, 41, 42] |
| **Define the problem** | 33.3 (6) [21, 36, 39, 40, 42, 43] |
| **Define expected outcomes** | 33.3 (6) [1, 13, 14, 37, 39, 43] |
| **Validate ToC** | 22.2 (4) [1, 21, 24, 42] |
| **Revise ToC** | 22.2(4) [36–38, 40] |

Source: authors' elaboration.

Table 5, below, presents these stages with further details. In addition to categorizing the stages for developing a ToC, we also classified the extracted data in terms of the "activities" conducted. While the classification in stages reflects the goal or purpose of each step taken, the classification into the activities summarizes the concrete actions undertaken to achieve this goal. These activities are depicted in the second column of Table 4. In the third column of this table, "Activities' details", we paraphrase the steps described in the included studies to present guidance on how to undertake each activity. The original material can be found in S3 Appendix.

We emphasize that among the "Activities", the lines "Identify and synthesize evidence base" refer to the processes of searching, selecting, producing, and synthesizing evidence for the corresponding stages of the ToC development. In these cases, the use of the term evidence is encompassing both tacit or colloquial evidence and explicit or published evidence, as further detailed in Table 5.

## Use of research evidence in the development and revision of a ToC

All included studies mentioned at least one source of evidence used to inform the development of the ToC. However, generally, the studies did not provide a structured and detailed discussion of how the incorporation of evidence was undertaken. Evidence sources included: literature reviews [1, 19, 20, 23, 36, 37, 42, 43], either systematic reviews or narrative reviews; review of reports and normative documents; surveys, interviews, meetings, workshops and focus groups with key actors, including community members and stakeholders [1, 13, 18, 20, 23, 24, 36, 38–40]; previous empirical studies; existing related psychological and sociological theories [1, 21, 37]; consultation with experts [18–20]; and previous theories of change [21, 22, 41]. Many studies [18, 20, 23, 36, 37, 39] combined more than one evidence source for the development of the ToC; such as a combination of literature review, consultations with stakeholders, and previous empirical studies.

The ToC diagram can be developed in a meeting with stakeholders, often in a workshop format with the help of a trained facilitator [13]. However, it is common to find the implementation or evaluation team making a first draft based on the literature review and then validating the draft in a stakeholder meeting. Mayne and Johnson (2015) [38] note that while bringing a pre-drafted ToC may decrease the level of buy-in, it makes it easier to bring relevant research findings into it.

De Silva et al. (2014) [13] report that in their study, the development of the ToC was supported by a ToC expert. They also conducted brainstorming activities with key actors to gather information about expected outcomes. Afterward, a survey with stakeholders was carried out,

**Table 4. Synthesis of Stages in ToC development and their respective activities.**

| Stages in ToC Development | Activities | Activities' details |
|---|---|---|
| **Define the problem** | **Define problem** | Identify [43], understand and define24 the main problems and its causes [36, 43] |
| | **Identify and synthesize evidence base** | • Describe the main concepts related to the problem and their interrelationships [21]<br>• Perform a narrative summary of the theories [21] relevant to the problem understanding<br>• Consult key actors [27] about relevant contextual conditions that may influence the problem and its solution [40]. This can be done, for instance, through interviews [36]<br>• Perform literature review [42] and identify systematic reviews on topics related to the problem [39] |
| **Define expected outcomes** | **Set goals** | Identify the expected results, as well as the possible long-term impact [13,42], according to the proposed interventions [14] through an iterative process (in groups, with stakeholders) [13] |
| | **Identify and synthesize evidence base** | Identify relevant evidence<br>• through health information and data use frameworks [39]<br>• by an prior pilot trial or an earlier feasibility study [37]<br>• through systematic reviews [39] |
| | **Map causal chains** | Identify short-term, mid-term and long-term outcomes in a causal chain [1] |
| **Define interventions** | **Identify and synthesize evidence base** | • Consult stakeholders and members of the intervention team (program evaluators and implementers) [18]<br>• Search published and gray literature about the intervention topic [43], existing conceptual frameworks or theories [1]<br>• Review institutional documents about the intervention [18]<br>• Direct program observation [1]<br>• Perform literature review [36], systematic review [32], or evidence synthesis [21] on effective interventions [36,43]<br>• Perform interviews and discussions, surveys, and program observation [1] |
| | **Define and detail interventions** | • Identify and select interventions data that could help stakeholders involved generate the desired change [20, 43]<br>• Identify patterns for grouping similar interventions [31]<br>• Identify characteristics of the environment which could influence the different links between intervention and programme outputs and outcomes, including socio-cultural, physical and personal factors [1]<br>• Identify barriers through search for existing health information and data use frameworks, as well as systematic reviews [32] and the interventions needed to overcome these barriers along the way [19]<br>• Identify the gaps in the research [19]<br>• Raise hypotheses (intervention mechanisms and contextual facilitators) and clarify how the changes will be implemented: develop, test and adapt the intervention program in question, consider possible implications of the findings for public policy [36] |
| **Define change mechanisms** | **Identify and synthesize evidence base** | Gather local evidence in order to refine "context-mechanism-outcome" hypotheses (CMO) [1]:<br>• Identify and synthesize the elements that will bring about change, as well as the intended outcomes deriving from them. Always based on findings from the previously conducted evidence identification and discussions with the advisory group [36] |
| | **Map causal chains** | Based on consultative approaches, add any assumptions or rationale to the links of the causal chain in the ToC [13]. Activities could include:<br>• Reflect on the collected contributions [13]<br>• Raise, discuss and register hypothetical connections between interventions (empirically examine the mediation between hypothetical interventions) [1]<br>• Raise, discuss and register hypothetical contextual barriers and facilitators [1]<br>• Interrelate the effects of the interventions and the evidence supporting each of them [14]<br>• Categorize theories among upstream (organizational), downstream (individual behavior) and medial (in between) [21]<br>• Identify assumptions, commonalities and differences [21]<br>• Define assumptions linking activities and outcomes [14]<br>• Understand the mechanisms of change [42] |

*(Continued)*

**Table 4.** (Continued)

| Stages in ToC Development | Activities | Activities' details |
|---|---|---|
| **Model ToC** | **Map causal chains** | Design the ToC including the following elements:<br>• intervention components [18]<br>• primary outcomes related to the intervention (each component of the intervention can be related to a causal outcome) 18<br>• contextual factors expected to modify the previous variables [18]<br>• information about the assumed causal connections between variables [18]<br>• identification of which causal connections are thought to be of great importance for the intervention's success [18]<br>• the specific interventions that need to happen to move from one outcome to the next (map in the ToC the specific intervention components needed to achieve each outcome) [13]<br><br>Practical considerations:<br>• Conduct a conceptual mapping to identify themes from the studies that could be used for the ToC model (the ToC map shows the multiple causal pathways through which the outcomes and activities work to achieve the desired impact) [19]<br>• Develop the theory through design intervention and creating realistic expectations [20]<br>• Represent, in an understandable way, the interaction between the intervention components, mechanisms and outcomes (consider using a software to help) [20]<br>• Map the intermediate outcome framework or causal pathway, without getting trapped into thinking about the specific intervention components that you think you will use, as this restricts your thinking regarding what is needed to achieve the desired impact [13]<br>• Ensure that each of the ToC components (reach, capacity changes, behavioral change, direct benefits, livelihood change) are identified as building blocks of a ToC [38]<br>• Development of a logic model for the theory of change based on inputs from workshops with stakeholders [42]<br>• Identify resources available [42]<br>• Develop a priority list of activities to be implemented [42]<br>• Develop an action plan [42]<br>• Conduct a pilot of intervention activities [42]<br>• Promote discussions with stakeholders [42]<br>• Backward map and connect the preconditions to achieve goals [43] |
| | **Draw a ToC** | ToC building process can start with either a blank page and a facilitator workshop with stakeholders, or with a ToC draft as a starting point and a discussion with stakeholders, ideally bringing prior research and evaluation findings into de ToC [38]<br>• Refine the ToC based on the results of the evaluation and stakeholder workshops [23]<br>• Integrate categorized findings, in order to integrate theoretical insights [21]<br>• Diagrammatically synthesize theories of change based on studies ToC reports and diagrams [22]<br>• Create a new ToC synthesizing the main elements of each ToC group [22] |
| **Validate ToC** | **Identify and synthesize evidence base** | Consult stakeholders:<br>• through focus groups of experts and the target population [20, 42]; through the Delphi method to reach consensus [28] or through workshops, discussing and validating the interventions [42]<br>• present a draft ToC based on different existing sources of information (theoretical models, frameworks and systematic reviews), explaining which elements of the ToC are evidence-informed and which are not [1]<br>• circulate this draft to program implementers and/or wider stakeholders for revision [24] |

*(Continued)*

**Table 4.** (Continued)

| Stages in ToC Development | Activities | Activities' details |
|---|---|---|
| **Revise ToC** | **Define monitoring and evaluation strategy** | Define indicators of success for each of the intermediate outcomes to evaluate whether every stage of the pathway leads to the final impact [19]<br>• Include indicators to measure each intervention level (e.g. patient, community, stakeholders and care providers) [13]<br>• For each intermediate outcome, choose at least one indicator to measure whether that intermediate outcome has been achieved [13]<br>• Decide on how each indicator will be measured and by whom (evaluation methods) [13]<br>• Establish when to measure each intermediate outcome with a rationale structure [13]<br>• Conduct a case study to evaluate the intervention performance [23] |
| | **Review ToC** | Theory of change should be seen as a process, evolving over time as more insight is gained [38]<br>• Collect sufficient evidence of effectiveness to justify rigorous evaluation/implementation (Step 5 from 6SQuID36) [36]<br>• Test hypotheses via quantitative analyses of effect mediation (to examine mechanisms) and moderation (to examine contextual contingencies). The approach of grounding these in hypothesis testing minimizes risks and ensures transparency [1]<br>• Adopt participatory approaches that foster and strengthen engagement and involvement of people–creating dialogue, inclusive opportunities to plan and decide on actions required [40]<br>• Agree and be clear on the purpose of planned actions–e.g. to develop a health assets model [40]<br>• Undertake a review/mapping of resources, including knowledge, skills, relationships etc. that will boost adoption of purposeful asset based approaches [40]<br>• Reframe existing relationships and the use of resources toward the purpose agreed in the planning phase [40]<br>• Introduce asset based approaches to reframe dialogue, planning and action with those already engaged [40]<br>• Use the perspectives and knowledge gained mapping assets to:<br>• Understand current health assets in place. Identify the location of assets–in neighborhoods, communities, organizations, etc. [40]<br>• Build on the range of assets available to support action. These actions can then be part of the mobilization phase of the ToC, using existing assets to support the agreed direction of development [40] |

Source: authors' elaboration.

**Table 5. Stages for ToC development.**

| Stages to develop a ToC | |
| --- | --- |
| Stage 1 | Define the problem |
| Stage 2 | Define expected outcomes |
| Stage 3 | Define interventions |
| Stage 4 | Define change mechanisms |
| Stage 5 | Model ToC |
| Stage 6 | Validate ToC |
| Stage 7 | Revise ToC |

Source: authors' elaboration.

a workshop was set up to foster debate, and, lastly, the ToC was further refined through another group session.

Few studies reported on how to use evidence to revise a ToC. The ToC was revised both in the initial development process, by experts [1, 19, 20] or peer review [1], and after the intervention was initiated, in response to evidence gathered by a discussion with practitioners [40], hypothesis testing (mechanisms and moderators evaluations) [37] and program evaluation [36].

**Challenges and barriers reported by the included studies.** The limitations reported in the included publications largely focused on the ToC development process and use of the ToC, the interventions performed, and the methods used in the studies.

Regarding the ToC, the authors reported that the process of using a ToC framework is time-intensive and it can lead to the oversimplification of complex relationships [1]. Moreover, ToCs are specific for each program and it is not simple to develop a ToC that is generalizable to other settings [41, 42]. The authors also reported that stakeholders' difficulty understanding the scientific method was a limitation to the evaluation of interventions using a ToC approach [1]. Finally, one scoping review highlighted that many studies do not report how the ToC used in the paper was developed [24].

Limitations related to the interventions performed in the studies were also mentioned, such as lack of time for a follow-up evaluation of the interventions [42], and limited availability and lack of evidence of culturally adapted interventions for LMICs [19]. In addition, some studies did not address adverse structural conditions that affect participants [36], or the effectiveness of interventions in the real world [1]. In another study, the authors reported problems in the intervention development process because it was not validated and relevant parameters were not included in the theory [20].

Regarding the methodological aspects of the studies, limitations reported by the systematic reviews included methodological flaws in the primary studies of relevant interventions, which may have affected statistical significance [14]. Insufficient quality assessment of the included studies [41]; absence of dual data extraction, and inability to effectively assess the quality of included articles [23]; and inclusion of studies in English only [14] were also reported as key limitations. In the only scoping review included, authors reported difficulties during the gray literature search, which was attributed to limitations of the Google search platform (insufficient database to capture gray literature studies) [24]. Regarding the other studies included, Yearwood et al. (2018) [43] highlighted that it was not possible to systematically evaluate the results achieved by the initiative, making it difficult to make precise conclusions about the effectiveness of the intervention.

**Gaps reported by the included studies.** The gaps reported in the included studies were the need for better methodological and reporting quality to assess the reliability of results [21, 22, 39, 40], the need to conduct evaluations of the proposed ToC models [19], and the need to synthesize previous theoretical frameworks to avoid possible biases [41]. In addition, studies suggest that ToC reports should be more detailed, especially regarding how the ToC was developed. The inclusion of a visual representation of the ToC (diagrams) would also better facilitate the reader's understanding and ease reproducibility [23, 24].

## Discussion

### Summary of main findings

This rapid review identified 18 studies that addressed the use of evidence in ToC or in the revise of a ToC in the health sector. This study adds to the existing literature in two ways. First, it provides an up-to-date and comprehensive review of what are the existing methods and recommendations for incorporating evidence into ToC frameworks in the health sector. Second, it creates a typology to classify the findings from the literature. In this typology, we propose a sequence of seven stages relevant to the process of evidence incorporation into ToCs.

An initial challenge for this review was the scarcity of data. In line with the conclusion from Breuer et al. (2016) [23] and Lam et al. (2021) [24], we find that authors rarely detail how the ToCs were developed and even more rarely include explicit considerations about evidence incorporation in the process. Future research projects should consider using the Breuer et al. (2016) [23] proposed checklist for reporting ToC in Public Health Interventions. It has five criteria, assessing the ToC approach, the ToC development process, presentation of the ToC, description of the intervention process, and use of the ToC for evaluating a policy. This checklist, however, has not yet been validated and might not be appropriate for all possible uses of a ToC. As such, it may require adaptations depending on the context in which it is used.

The studies included in this review, nevertheless, provide a rich portrayal of the field. The findings corroborate the definition of ToC provided by De Silva et al. (2014) [13], according to which a ToC graphically presents how and why an intervention is expected to work and, in general, is developed by a participatory process [4].

We observed significant diversity across topics and areas within the health sector in which ToC approaches can be applied. It was also noted that there is no single way to develop a ToC that suits all purposes, but that adaptations are made according to the context of its application.

### Stages for ToC development

The stages for developing a ToC were observed to be different in the included studies. The process of developing a ToC has multiple possibilities regarding its form and content, being essentially dynamic and nonlinear. However, for didactic and visual purposes, we classified and organized these stages in a logical order (See Table 5, below).

It is important to highlight that this logical sequence is our interpretation, based on a qualitative analysis of the extracted data. As we mentioned before, the authors often do not provide a structured presentation of how the ToCs were developed. Moreover the stages are rarely followed in this strict logical sequence.

Prior to the stages, it is also important to discuss the purpose and goal of the ToC. As we mentioned above, ToCs can have many purposes, such as supporting the implementation of an intervention, supporting monitoring and evaluation efforts, sharing and aligning knowledge and expectations about outcomes among stakeholders, synthesizing knowledge and communicating to a specific audience. Given the time, work, and coordination requirements to

gather stakeholders and research evidence to develop a ToC, it is paramount that its goals are well defined beforehand.

## Methods for evidence incorporation into ToC development

From Table 4 above, it is possible to observe how different sources of evidence are used in the development of ToCs. There were three main sources of evidence: literature search, consultation with key actors, and use of institutional data. More specifically, there were a variety of ways to use evidence in ToC, such as searching for systematic reviews, conducting systematic and narrative literature reviews, using primary studies, using theoretical frameworks, using institutional and epidemiological data, conducting meetings, interviews, surveys, and workshops with stakeholders, or a combination of these components. Therefore, both colloquial and scientific evidence were commonly used sources of knowledge to support ToCs. Four studies reported the use of evidence for revising a ToC. The findings showed that revising a ToC can be done by evaluating programs, mechanisms, and moderators, and by consultation with experts and other key actors.

In Table 6, below, we synthesize the methods for evidence identification and synthesis presented in the Results section (see Table 4). These methods are organized by ToC development stage:

Table 6 describes what methods for evidence incorporation the included studies applied for the different stages of ToC development, with the stage "Define expected outcomes" supported, mostly, by the use of scientific evidence, both at a global and local level. As for the "Validate ToC" stage, we identified the use of evidence only at the local level, but, in a way, incorporating both scientific and colloquial evidence, through experts and users. We also observe that for the "Define the problem", "Define intervention" and "Revise ToC" stages,

**Table 6. Methods used by the included studies for evidence incorporation.**

| Stages of ToC development | Methods for evidence identification | | |
| --- | --- | --- | --- |
| | Use of Institutional data | Literature search | Stakeholders consultation |
| **Define the problem** | • Search for information available in the health system | • Literature review<br>• Search for systematic reviews | • Interviews |
| **Define expected outcomes** | • Prior feasibility study or a pilot study<br>• Search for information available in the health system | • Search for systematic reviews | • Focus Groups |
| **Define interventions** | • Program observation | • Literature review<br>• Search in published and gray literature<br>• Search in existing conceptual frameworks or theories | • Interviews<br>• Debates<br>• Surveys |
| **Define change mechanisms** | *Not discussed* | *Not discussed* | • Consultations |
| **Model ToC** | *Not discussed* | • Syntheses of previous theories of change | • Workshops |
| **Validate ToC** | *Not discussed* | *Not discussed* | • Focus groups of experts and users<br>• Delphi method to reach consensus |
| **Revise ToC** | • Program evaluation<br>• Mechanisms and moderators evaluation | *Not discussed* | • Expert consultation<br>• Peer review |

Source: authors' elaboration.

authors explore evidence both at a global and local level, including scientific and colloquial knowledge. This demonstrates the ability of a ToC to be built from different sources of knowledge, considering colloquial evidence without giving up on scientific rigor.

Table 6, however, should not be understood as a recommendation of which methods should be applied for evidence incorporation into ToCs, nor as a conceptually informed exhaustive list of the possibilities in that regard. The included studies did not present justification as to why a specific method was chosen, or why other methods were left out. Moreover, no study provided a normative account on how to incorporate evidence into ToC development, nor provided specific guidance for that purpose.

Therefore, we identify a gap in the existing literature. The fact that only 18 studies explicitly discuss how to incorporate evidence into ToC development in the health sector, and none offers guidance or a framework for this shows that it is a neglected area. This stands in sharp contrast with the increasing recognition of evidence as an indispensable element for policy-making in the health sector.

This literature gap is in line with the findings from Lam et al. (2021) [24], which report a fragmentation in the field and a lack of guidance and conceptual consistency regarding how to develop a ToC. There is, therefore, an existing need for a new theoretical framework, offering clear advice on how to incorporate evidence into ToC development. This framework should draw on the existing practice, as discussed in this rapid review, while also integrating methodological and conceptual considerations, to guide the many ways through which evidence can be used to inform ToCs.

### Strengths and limitations of the present study

This project applied a rapid review methodology, following the guidance provided by Tricco et al. (2017) [27]. Regardless of the abbreviated approach, this endeavor respected the scientific method and the key principles of evidence synthesis, including rigor, transparency, and reproducibility. Furthermore, the shortcuts adopted in this review allowed us to focus on the most critical and relevant literature to our research questions.

A few adaptations from a complete systematic review were made. First, only studies involving the health sector were included in the eligibility criteria, which may have limited relevant information about ToCs. Second, the inclusion criteria for systematic reviews, scoping reviews, and methodological studies and guidelines focusing on the development of theories of change in the health sector may not have covered the entire literature on the use of evidence in the theory of change. Third, only studies in English, Spanish, and Portuguese were included, due to the language skills of the reviewers. Fourth, data extraction and the methodological quality assessment of the studies were conducted by one reviewer and checked by another. Fifth, some of the included studies did not explicitly state their research design, which required the reviewers to classify these studies as a specific type to choose and apply the methodological assessment tools that were the most adequate.

### Conclusion

This rapid review identified studies that addressed the use of evidence in ToC, as well as strategies for developing a ToC in healthcare contexts. Several strategies for using evidence in the development of a ToC were reported, such as retrieving evidence from the scientific literature, using theoretical frameworks, and consultations with stakeholders. These strategies were applied in isolation or combined. Little evidence was found about the use of evidence for the revision and updating of a ToC. No study provided a normative account on how to incorporate evidence into ToC development, nor provided specific guidance for that purpose.

Furthermore, we identified a significant gap between the available literature on evidence integration for ToCs and the state-of-the-art debates around Evidence-Informed Decision-making (EIDM). While the EIDM debate has been advocating for increased use of evidence to support all stages of the policy cycle [48], it seems that ToC development has yet to be fully addressed. Similarly, the literature specializing in ToCs does not seem to incorporate many of the tools developed within the EIDM area, such as Rapid Reviews, Scoping Reviews, Evidence Gap Maps, among others. It also does not provide a structured account of how to bridge the techniques for ToC development with the need to systematically and transparently incorporate evidence into all stages of the policymaking process. A possible way of solving this issue could be the establishment of a protocol that ensures that different approaches share fundamental elements concerning the use of evidence in the development of a ToC. Therefore, it is crucial that future research addresses this gap.

Based on the findings, we conclude that there is a need for studies to explore in a clearer and more detailed way how evidence can be used in the development and updating of ToCs in the health sector. It is also important that studies that report a ToC should explicitly outline the process through which it was developed.

## Supporting information

**S1 Appendix. Search strategies.**
(DOCX)

**S2 Appendix. Methodological quality of the included studies.** *Critical domains; CL: Critically low.
(DOCX)

**S3 Appendix. Stages of ToC development: Qualitative analysis.**
(DOCX)

**S4 Appendix. Full-text articles excluded, with reasons.**
(DOCX)

**S5 Appendix. Characteristics of the included studies.** *Study type: When the authors did not inform the study design, the reviewers attributed a classification based on the provided description of the methods. HIC: High-income economies; LMIC: Lower-middle-income economies; UMIC: Upper-middle-income economies. ** The systematic review had a ToC as one of its outputs.
(DOCX)

**S6 Appendix. Results of the included studies.**
(DOCX)

**S1 Checklist. PRISMA 2020 checklist.**
(DOCX)

## Acknowledgments

We thank the following individuals who assisted the implementation of this project: Laura dos Santos Boeira and Marcel Henrique de Carvalho.

## Author Contributions

**Conceptualization:** Peter Nichols DeMaio, Tanja Kuchenmüller.

**Data curation:** Cecilia Setti, Leo Heikiti Maeda Arruda, Roberta Crevelário de Melo, Bruna Carolina de Araujo.

**Formal analysis:** Davi Mamblona Marques Romão, Cecilia Setti, Leo Heikiti Maeda Arruda, Roberta Crevelário de Melo, Bruna Carolina de Araujo.

**Funding acquisition:** Davi Mamblona Marques Romão.

**Investigation:** Cecilia Setti, Leo Heikiti Maeda Arruda, Roberta Crevelário de Melo, Bruna Carolina de Araujo.

**Project administration:** Davi Mamblona Marques Romão.

**Supervision:** Davi Mamblona Marques Romão, Cecilia Setti.

**Validation:** Davi Mamblona Marques Romão, Cecilia Setti, Leo Heikiti Maeda Arruda, Roberta Crevelário de Melo, Bruna Carolina de Araujo, Audrey R. Tan, Peter Nichols DeMaio, Tanja Kuchenmüller.

**Visualization:** Davi Mamblona Marques Romão, Cecilia Setti.

**Writing – original draft:** Davi Mamblona Marques Romão, Cecilia Setti, Leo Heikiti Maeda Arruda, Roberta Crevelário de Melo, Bruna Carolina de Araujo.

**Writing – review & editing:** Davi Mamblona Marques Romão, Cecilia Setti, Leo Heikiti Maeda Arruda, Audrey R. Tan, Peter Nichols DeMaio, Tanja Kuchenmüller.

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
