## [Decision Letter · Decision Letter 0]

8 Nov 2022

PONE-D-22-21768

Integration of evidence into Theory of Change frameworks in the healthcare sector: a rapid systematic review

PLOS ONE

Dear Dr. Romão,

Thank you for submitting your manuscript to PLOS ONE. After careful consideration, we feel that it has merit but does not fully meet PLOS ONE’s publication criteria as it currently stands. Therefore, we invite you to submit a revised version of the manuscript that addresses the points raised during the review process.

We look forward to receiving your revised manuscript.

Kind regards,

Muhammad Shahzad Aslam, Ph.D.,M.Phil., Pharm-D

Academic Editor

PLOS ONE

Journal Requirements:

Reviewers' comments:

Reviewer's Responses to Questions

**Comments to the Author**

1. Is the manuscript technically sound, and do the data support the conclusions?

Reviewer #1: Yes

Reviewer #2: Yes

2. Has the statistical analysis been performed appropriately and rigorously? 

Reviewer #1: N/A

Reviewer #2: N/A

3. Have the authors made all data underlying the findings in their manuscript fully available?

Reviewer #1: Yes

Reviewer #2: Yes

4. Is the manuscript presented in an intelligible fashion and written in standard English?

Reviewer #1: Yes

Reviewer #2: Yes

5. Review Comments to the Author

Reviewer #1: This review is clearly written, well organised and reported the strategies using research evidence when developing or adapting the 'Theory of Change' in healthcare context. This review also proposed the stages for 'Theory of Change development. A good dept and explanation in the results and discussion sections by the authors.

Reviewer #2: 1. There is a reference to the "Theory of Change" on line 52 (the first line of the article). Prior to that, authors should define and explain what "the theory of change" is, in simple and easy words. Describe what you mean by "change" in this context, how the "theory of change" came to be, and give a background on the theory of change.

2. Authors should explain how the ToCs were developed.

3. It might be helpful to define some of these lesser-known terms like ToC, logframes and logic model.

4. In line 97 change the word "ex-post" or explain it (based on analysis of past performance).

5. There was no explanation provided by authors regarding the word "health sector" before objectives. Authors should provide information about how ToC is used in the health sector. If there is no literature available or none of the recent studies has systematically investigated the use of evidence to inform ToCs in the health sector, then how would you conduct a systematic review in the health sector?

6. The authors did not explain the rapid review.

7. In line no 148, the authors stated that "This rapid review adds to the existing literature" as "it provides an up-to-date and comprehensive review of the existing methods". Authors should discuss those existing methods first and then justify this statement.

8. In line no 148, "we propose a sequence of 7 stages relevant to the process of evidence incorporation into ToCs". How you came up with these 7 stages. The authors should provide a description of the 7 stage process and how it came to be.

9. In line no 172, "Given the above-mentioned constraints, some adaptations were made to adjust to the needs" kindly provide the line number where the authors mentioned those constraints above.

10. In line no 196, please indicate the starting and ending date of articles searched. Or did authors search only in a day?

11. In line 199, the authors mention a topic expert. Who is that expert? What are the criteria for becoming an expert, and why only one expert? Also why additional search for gray literature was not conducted?

12. In lines no 297-299, the authors mentioned studies with high-income countries, low and middle-income countries and high-income economies "Seven studies were carried out in countries 298 that, according to The World Bank (2021) [37], are classified as high-income economies (HIE) 299 [10,12,17,27,29,30,35], three in Lower-middle-income economies (LMIE) [5,10,27]". In this sentence reference number 10 is mentioned in both high-income economies and lower-medium-income economies. Aggarwal, S., Patton, G., Berk, M. & Patel (reference 10) is only represented in LMIE. Also, check the reference no 27.

13. Kindly check reference no 30 in line no 309. In my opinion, this reference does not match the criteria.

6. PLOS authors have the option to publish the peer review history of their article (what does this mean?). If published, this will include your full peer review and any attached files.

Reviewer #1: **Yes: **Dr Rimah Melati Ab. Ghani

Reviewer #2: **Yes: **Saima Nisar

---

## [Author Response · Author response to Decision Letter 0]

12 Dec 2022

Dear Reviewers,

We appreciate your time reviewing our paper and providing valuable feedback. Your careful reading and insightful comments led to improvements in the current version. We carefully considered the comments and tried our best to address all of them. We hope the new manuscript meets your standards. We welcome further constructive comments, if necessary. In the appendix "Response to reviewers" you can find the point-by-point responses, that are also visible with track changes, in the appendix "Revised Manuscript with Track Changes".

Kind regards,

Davi Romão

---

## [Decision Letter · Decision Letter 1]

4 Jan 2023

PONE-D-22-21768R1Integration of evidence into Theory of Change frameworks in the healthcare sector: a rapid systematic reviewPLOS ONE

Dear,

Thank you for submitting your manuscript to PLOS ONE. After careful consideration, we feel that it has merit but does not fully meet PLOS ONE’s publication criteria as it currently stands. Therefore, we invite you to submit a revised version of the manuscript that addresses the points raised during the review process.

We look forward to receiving your revised manuscript.

Kind regards,

Muhammad Shahzad Aslam, Ph.D.,M.Phil., Pharm-D

Academic Editor

PLOS ONE

Reviewers' comments:

Reviewer's Responses to Questions

**Comments to the Author**

1. If the authors have adequately addressed your comments raised in a previous round of review and you feel that this manuscript is now acceptable for publication, you may indicate that here to bypass the “Comments to the Author” section, enter your conflict of interest statement in the “Confidential to Editor” section, and submit your "Accept" recommendation.

Reviewer #2: All comments have been addressed

2. Is the manuscript technically sound, and do the data support the conclusions?

Reviewer #2: Yes

3. Has the statistical analysis been performed appropriately and rigorously? 

Reviewer #2: N/A

4. Have the authors made all data underlying the findings in their manuscript fully available?

Reviewer #2: Yes

5. Is the manuscript presented in an intelligible fashion and written in standard English?

Reviewer #2: Yes

6. Review Comments to the Author

Reviewer #2: 1. In line 70, "The concept of ToC is related to, and often overlaps with, several other tools and frameworks." Authors should mention the other tools and frameworks' names first with this sentence and then explain them.

2. The authors mentioned different line numbers in the "Authors' replies to reviewers' responses" and the corrections done on different lines in the article. This confusion made the review process difficult.

3. In line 200, kindly explain the "rapid review" with references.

4. In lines 205 and 206, authors repeated the same sentence: "We used a rapid review approach to be able to provide high quality evidence for decision making, in a timely way."

5. In the previous comment, "comment no. 7," the authors were asked to discuss those existing methods first and then justify this statement. However, the authors only stated that "the majority of the existing literature on the topic focuses on the development of a ToC but only provides brief descriptions of the methods used to incorporate evidence into this process."The authors did not explain the existing methods. 

6. In the previous comment, "comment no. 8," the authors were asked, "How did you come up with these 7 stages?" The authors' reply is not satisfactory. Authors should explain the details of 7 stages before applying them.

7. PLOS authors have the option to publish the peer review history of their article (what does this mean?). If published, this will include your full peer review and any attached files.

Reviewer #2: **Yes: **Saima Nisar

---

## [Author Response · Author response to Decision Letter 1]

2 Feb 2023

Dear Reviewer,

We appreciate your time reviewing our paper and providing valuable feedback. As before, we have carefully considered the comments and tried our best to address all of them. We hope the new manuscript meets your standards. The authors welcome further constructive comments, if necessary. 

In the appendix "Response to reviewers" you can find the point-by-point responses, that are also visible with track changes, in the appendix "Revised Manuscript with Track Changes".

Sincerely,

The Authors.

---

## [Decision Letter · Decision Letter 2]

23 Feb 2023

Integration of evidence into Theory of Change frameworks in the healthcare sector: a rapid systematic review

PONE-D-22-21768R2

Dear Dr. Romão,

We’re pleased to inform you that your manuscript has been judged scientifically suitable for publication and will be formally accepted for publication once it meets all outstanding technical requirements.

Kind regards,

Muhammad Shahzad Aslam, Ph.D.,M.Phil., Pharm-D

Academic Editor

PLOS ONE

Additional Editor Comments (optional):

Reviewers' comments:

Reviewer's Responses to Questions

**Comments to the Author**

1. If the authors have adequately addressed your comments raised in a previous round of review and you feel that this manuscript is now acceptable for publication, you may indicate that here to bypass the “Comments to the Author” section, enter your conflict of interest statement in the “Confidential to Editor” section, and submit your "Accept" recommendation.

Reviewer #2: All comments have been addressed

2. Is the manuscript technically sound, and do the data support the conclusions?

Reviewer #2: Yes

3. Has the statistical analysis been performed appropriately and rigorously? 

Reviewer #2: N/A

4. Have the authors made all data underlying the findings in their manuscript fully available?

Reviewer #2: Yes

5. Is the manuscript presented in an intelligible fashion and written in standard English?

Reviewer #2: Yes

6. Review Comments to the Author

Reviewer #2: (No Response)

7. PLOS authors have the option to publish the peer review history of their article (what does this mean?). If published, this will include your full peer review and any attached files.

Reviewer #2: **Yes: **Saima Nisar

---

## [Editor Report · Acceptance letter]

28 Feb 2023

PONE-D-22-21768R2 

Integration of evidence into Theory of Change frameworks in the healthcare sector: a rapid systematic review 

Dear Dr. Romão:

I'm pleased to inform you that your manuscript has been deemed suitable for publication in PLOS ONE. Congratulations! Your manuscript is now with our production department. 

Kind regards, 

on behalf of

Dr. Muhammad Shahzad Aslam 

Academic Editor

PLOS ONE